# Association between Smartphone Addiction and Suicide

**DOI:** 10.3390/ijerph191811600

**Published:** 2022-09-15

**Authors:** Oyuntuya Shinetsetseg, Yun Hwa Jung, Yu Shin Park, Eun-Cheol Park, Suk-Yong Jang

**Affiliations:** 1Department of Public Health, Graduate School, Yonsei University, Seoul 03722, Korea; 2Institute of Health Services Research, Yonsei University, Seoul 03722, Korea; 3Department of Preventive Medicine, Yonsei University College of Medicine, Seoul 03722, Korea; 4Department of Healthcare Management, Graduate School of Public Health, Yonsei University, Seoul 03722, Korea

**Keywords:** smartphone addiction, smartphone over-dependence, internet addiction, cell phone, addiction, suicide ideation, suicide attempt, self-harm, depression, adolescence mental health

## Abstract

Objective: This study aimed to explore the associations between smartphone overdependence (smartphone addiction) and suicidal ideation and attempts among Korean adolescents to acknowledge the risk of smartphone overuse. Methods: Data were obtained from the results of the 2020 Korea Youth Risk Behavior Web-based Survey. Smartphone addiction was categorized into three groups: adolescents who scored less than 23 were categorized as the general user group and formed the reference, those with scores of 23–30 were categorized as the potential-risk user group, and those with scores higher than 31 were categorized under the high-risk user group. Suicidal ideation and suicide attempt were identified as dependent variables in the present study. Multiple logistic regression was used to analyze the relationship between SA and suicidal ideation and suicide attempt status among Korean adolescents. Results: This study included 41,173 general users of smartphones, 12,142 potential-risk users, and 1633 high-risk users from among 54,948 adolescents who were middle- and high-school students. Adolescents with potentially risky smartphone use showed a higher likelihood of suicidal ideation (OR: 1.50, CI: 1.42–1.60). Similarly, adolescents with high-risk smartphone use showed a significant risk of suicidal ideation (OR: 2.49, CI: 2.21–2.81) and suicide attempt (OR: 1.87, CI: 1.48–2.37) compared to the adolescents who were general users. Conclusion: Our study results encourage parents and social workers to acknowledge that adolescents’ smartphone addiction leads to a higher risk to their mental health, wherein they may engage in suicidal ideation and even resort to a suicide attempt.

## 1. Introduction

Suicide is the second leading cause of death among adolescents globally [1]. Comparing the age-standardized suicide rate among Organization for Economic Co-operation and Development (OECD) countries (OECD standard population per 100,000), Korea recorded the highest level at 24.6 (as of 2019) compared to the OECD average of 11 [2]. Since 2011, suicide has been the leading cause of death among young people in Korea from adolescents up to those in their 30s [3]. The suicide rate has been gradually increasing year by year in the last two decades, which adds to the global burden of disease.

Nowadays, smartphones have become an essential and basic necessity of our lives. Nevertheless, they also have several negative effects on adolescents in their growing ages. There are many studies [4,5,6,7,8,9] reporting that adolescents’ smartphone overuse is associated with their self-esteem, life stress, anxiety, depression, impulsivity, friends and social communication conflict, and family communication and conflict. Further, smartphone addiction and becoming “hikikomori” at a young age are becoming a significant problem during the COVID-19 pandemic [4,5,6,7,8,9]. Thus, adolescents’ smartphone addiction is a worldwide problem that humans are facing, but because of the increasingly essential nature of smartphones, families are unable to directly restrict its use. In addition, parents have not fully understood and realized the serious harm of smartphone overuse for their children.

Internet overuse is also a manifestation of Internet addiction in a broad sense, depending not only on the social use of the Web but also on gambling and online shopping on the Internet and Internet porn addiction. The drama of the latter type of addiction lies in the fact that adolescents are in a transition period, which overtakes them during puberty, and then adolescents develop antisocial ideas about sex, which can become ingrained and remain as a trace that may have a harmful enduring effect [10,11,12]. During their sensitive period of growth, adolescents are both physically and psychologically immature. Previous research has reported that the physical processes that make adolescents more prone to poor decision-making and cause difficulty in emotional regulation and self-control are because of structural complexity and the imbalance of structural and functional changes between the subcortical areas generated by emotion- and pleasure-related experiences. A possible mechanism for the above phenomenon is that the prefrontal cortex area that is responsible for the top–down regulation of emotions and impulses has not reached its complete maturity by adolescence [13,14,15,16].

A national surveillance study on Internet addiction shows that Internet addiction is more harmful and dangerous than other forms such as alcoholism, smoking, and sexual addiction that are widespread among the population. Internet addiction contributes to depression, behavioral changes, social communication disorders, and anxiety, and the unreal virtual environment of the Internet often creates an abnormal psychological state that is alienated from real life, causing depression so that individuals cannot adapt to the real-life environment [17,18]. Furthermore, smartphone overuse has negative effects and introduces risks that are difficult for adolescents to handle, such as internet bullying, pornography, and low academic achievement [19,20]. The negative effects include poor peer and family relationships, vision loss, etc. [5,21]. Therefore, a variety of drastic physical, psychological, and social relation changes that adolescents have no control over and cannot avoid occur as a result of smartphone addiction [22]. The addiction could grow extremely hard to control, both mentally and physically; the resulting stressful events and experiences can cause adolescents to mentally break down, consider self-harm, have suicidal thoughts, and even resort to suicide attempts.

A few studies have found a correlation between smartphone overuse and self-harm and suicidal ideation [23,24,25,26,27]; many previous studies have also found related potential risk factors. These include depression, anxiety, impulsiveness, family related issues, sociodemographic status, academic stress, and substance use [6,8,28]. Moreover, men with high levels of internet addiction are at higher risk of suicide than women. It can be said that Internet addiction is a mediating variable in terms of gender [29,30]. Gaming was associated with Internet problems in men, but aggression and impulsivity showed associations with smartphone use in women [31]. However, there were no studies that directly explored the smartphone addiction scale to identify associations between smartphone addiction and both suicidal ideation and suicide attempts among adolescents. Therefore, the purpose of this study is to spread awareness among parents and social workers about the vicious circle of smartphone overuse and risk that can negatively impact adolescents’ lives. Further steps can be taken to prevent, moderate, and supervise the tendencies of adolescent smartphone use.

## 2. Methods

### 2.1. Study Participants

The data were obtained from the results of the Korea Centers for Disease Control and Prevention’s 2020 Korea Youth Risk Behavior Web-based Survey (KYRBWS), known as KYRBWS-XVI—an anonymous self-report online survey. This survey aimed to calculate statistics on the health behavior of Korean adolescents. It was used as basic data necessary for the planning and evaluation of health promotion projects. The 2020 (16th) survey explored 103 questions in 15 areas, including smoking, drinking, and physical activity, and drew 93 indicators. The survey was conducted with 793 schools (398 middle schools, 395 high schools) and 54,948 people participating in the survey due to Coronavirus disease 2019 (COVID-19), and the participation rate was 94.9–95.3% based on the number of students.

### 2.2. Suicidal Ideation and Suicide Attempt

The dependent variables of the present study were both suicidal ideation and suicide attempt. Suicidal ideation was inferred based on the question: “At any time within the last 12 months, did you think about trying to kill yourself?”; the responses were “Yes” and “No”. Suicide attempt was questioned by “At any time with in the last 12 months, did you ever attempt suicide?”, and the responses were one of the following: (1) “No, I never attempted suicide” or (2) “Yes, I have attempted suicide at least once”.

### 2.3. Smartphone Addiction

Smartphone addiction was determined by adolescents’ answers to statements regarding the ten items of the smartphone dependence screening tool, which was developed from the KYRBS’s self-reporting of the smartphone addiction scale. This scale had a total of 40 points answered on a four-point Likert scale, where 1 indicates a strong disagreement with the statement and 4 indicates a strong agreement with the statement. The ten items of the scale included (1) It fails every time you reduce the smartphone hours; (2) It is difficult to control the usage time of smartphone; (3) It is difficult to keep proper smartphone usage time; (4) It is hard to focus on other things if you have a smartphone next to you; (5) Thinking of smartphone does not leave your head; (6) I strongly desire to use smartphone; (7) I have problems with my health because of smartphone use; (8) I have a hard time with my family because of smartphone use; (9) Due to the use of smartphones, I have experienced severe conflicts in friends, colleagues, and social relationships; and (10) Due to the smartphone, there are difficulties in performing tasks (such as study or work). The smartphone addiction levels were ascertained based on the scores. Smartphone addiction was categorized into three groups: adolescents who scored less than 23 were categorized as the general user group for reference, those with scores of 23–30 were categorized as the potential-risk user group, and those with scores of higher than 31 were the high-risk user group [32]. Smartphone addiction categorization was according to the Korean Internet Addiction Prevention Center.

### 2.4. Covariates

Other confounding variables representing the study participants’ sociodemographic characteristics were gender, school grade (middle school, high school), school achievement (high, moderate, and low), subjective economic status (high, average, and low), and family living structure (living with family or without family). Health-related characteristics were sleeping duration for overcoming fatigue (enough, normal, not enough), consumption of fast food and physical activity (≤2 times/week, ≥3 times/week), tobacco use (Yes or No for the question: “Do have tobacco use experience?”) and alcohol consumption (response options were Yes or No) within a month, and alcohol drinking experience (in the last 30 days). When analyzing the relationship between smartphone addiction and suicidal ideation and suicide attempt across the study participants, all covariates were examined by the responses to the corresponding question.

### 2.5. Statistical Analysis

A chi-squared test was used to evaluate the basic characteristics of the participants. Multiple logistic regression analysis was performed to evaluate the association between smartphone addiction risk and suicidal ideation and attempt. Results included odds ratio (ORs), and 95% confidence intervals (CIs) were calculated. Results from analysis of variance and tests of independence were also valid. The trend test was conducted to identify the relationship between the independent and dependent variables. All analyses were conducted using SAS 9.4 (SAS Institute Inc., Cary, NC, USA). *p*-values lower than 0.05 were considered significant.

## 3. Results

This study included 54,948 adolescents belonging to middle and high school (middle school: 28,961; high school: 25,987) in South Korea. Of all participants, 41,173 (74.9%) were categorized as the general user group, 12,142 (22.1%) as the potential-risk group, and 1633 (3.0%) as the high-risk user group (Table 1). Girls’ scores exceeded boys in terms of potential and high risk of overusing smartphones (potential risk: 6903 (56.9%), high risk: 1040 (63.7%)).

Table 2 indicates the general characteristics of the participants. Adolescents who experienced suicidal ideation within the last 12 months included 5979 (10.9%) of 54,978 participants. By category, 3680 (6.7%) adolescents in the general user group, 1860 (3.4%) in the potential-risk group, and 439 (0.8%) in the high-risk group experienced suicidal ideation in the last 12 months.

Adolescents who had engaged in suicide attempt within the last 12 months included 1121 (2.0%) of 54,978 participants. By category, 721 (1.3%) adolescents in the general group, 305 (0.6%) in the potential-risk group, and 95 (0.2%) in the high-risk group had attempted suicide in the last 12 months.

Table 3 shows the results of multivariate logistic regression analysis for the factors associated between smartphone addiction risk and suicidal ideation and suicide attempt. Adolescents who had potential risk of smartphone addiction exhibited higher likelihood of suicidal ideation (OR: 1.50, CI: 1.41–1.60) and those who had high-risk of smartphone use exhibited significant likelihood of suicidal ideation (OR: 2.49, CI: 2.21–2.81) compared to the adolescents who had general use of smartphones. Adolescents who had high-risk of smartphone use showed increasing risk of suicide attempt (OR: 1.87, CI: 1.48–2.37) compared to those who had general use of smartphones.

Table 4 shows the results of subgroup analysis stratified by independent variables. Adolescents who had potential risk of smartphone addiction exhibited higher likelihood of suicidal ideation. Further, adolescents who had high risk of smartphone addiction exhibited increased risk of suicidal ideation and attempt across all sociodemographic, socioeconomic, and health-related variables compared to the general smartphone use group.

Table 5 displays logistic regression analysis results of association between individual components of smartphone addiction risk and suicidal ideation and attempt; adolescents who scored higher on statements such as, “Due to the use of smartphones, I have experienced severe conflicts in friends, colleagues, and social relationships” and “Due to the smartphone, there are difficulties in performing tasks (such as study or work)” exhibited significant likelihood of both suicidal ideation (OR: 2.08–2.42) and attempt compared to the general users (OR: 1.87–2.45).

## 4. Discussion

In this study, we identified significant associations between adolescents’ smartphone addiction and suicidal ideation and suicide attempt among Korean adolescents. Adolescents who had potential and high risk of smartphone addiction showed increasing likelihood of both suicidal ideation and suicide attempt. Adolescents who strongly agreed with the smartphone addiction scale’s eighth (“I have a hard time with my family because of smartphone use”) and ninth (“Due to the use of smartphones, I have experienced severe conflicts in friends, colleagues, and social relationships”) items showed significantly higher likelihood for both suicidal ideation and attempt. Adolescents’ overdependence on smartphones can lead to social isolation and even a breaking down of their relationship with their closest ones, which can ultimately lead to loneliness and mental breakdown.

In our results, the potential-risk and high-risk smartphone user groups had significant risk for suicidal ideation and suicide attempt, which indicates that smartphone overuse can have a negative effect on adolescent’s mental health, as evidenced in previous studies [6,33]. One study found that Internet addiction had significantly higher relation to neuroticism and psychoticism, which appeared as significantly higher scores on the Strengths and Difficulties Questionnaire (SDQ) subscales of emotional symptoms, conduct problems, and hyperactivity [34]. This may increase depression, anxiety, and mental health problems and cause a stronger link to suicidal ideation and suicide attempts.

Depressive symptoms and suicide are linked to time-related poor sleep, late-night use of digital devices, and smartphone addiction. Attention deficit and hyperactivity disorder (ADHD)-related behaviors include sleep problems and general screen time, which activates dopamine and reward pathways. Addictive screen time use is associated with reduced social distress and craving similar to subject affiliation behavior [35,36]. In addition, previous meta-analysis shows that Internet addiction was associated with increased suicidality after adjusting for potential confounding variables including depression [18].

In our results, high risk of smartphone addiction was associated with suicidal ideation and attempts (Table 4). A possible explanation is that the Internet provides and creates many activities that may be related to suicide problems. The results of this study suggest that online gaming, chatting, watching movies, shopping, and gambling are associated with suicide attempts. Online movies and chat rooms may discuss and share the suicide experience. Therefore, adolescents may learn about suicidal behaviors and learn about actual suicide methods [37,38,39].

Previous studies also state that adolescents who had poor relationships with their family and peers (social relationships) in their real life may establish relationships on the Internet or increase their smartphone use to escape such negative feelings. This may culminate in smartphone addiction [6,40,41,42,43]. Such an avoidance-based coping strategy can worsen the situation and make it hard to deal with life events and experiences, leading to no control of the stressful and depressing situation; adolescents may, therefore, experience mental breakdown, resulting in thoughts about suicide and even suicide attempts [44].

There are several limitations in this study. First, the data used in this study were cross-sectional and may be affected by reporting bias because they were collected through self-managed online surveys. This prevents us from inferring evidence of causality between the variables in this study. Second, our study was conducted in 2020. This was during the COVID-19 pandemic, and even though there was no lockdown in South Korea, public places such as restaurants, coffee shops, and movie theaters were restricted from operating or had restricted business hours. This may have affected the quality of life of young people in many ways.

Despite these limitations, this study had several strengths. First, we used the most recent and highest number of participants from a nationally representative database. Second, the data that we used were part of the smartphone addiction scale, which was constructed for the first time in 2021. Lastly, there were no published studies that directly examine smartphone addiction and both suicidal ideation and suicide attempt; our study bridged this gap.

In addition, future research may focus on other facts that lead adolescents to engage in smartphone overuse or suffer from addiction and can aim to understand the longitudinal impact of specific application of smartphone use on the overdependence behaviors of adolescents, which can prevent related mental health issues and help strengthen the current smartphone-use-related policies and guidelines.

## 5. Conclusions

Our study shows significant correlation between smartphone addiction risk level and suicidal ideation and suicide attempt among Korean adolescents. The study results suggest that it is important to encourage parents and social workers to acknowledge smartphone overuse or addiction as it has significant risks to adolescents’ mental health. Therefore, there is a persistent need for effective prevention plans and specific government policies that can reduce smartphone overuse and the suicide rate. First, smartphone-based tools can provide an effective way to prevent smartphone addiction and further risks, which is corroborated by certain studies that revealed that smartphone-based tools have reduced depressive symptoms and anxiety. Second, based on our results, we suggest that parental monitoring, emotional support, and moderate restriction of smartphone use are a few actions that can be taken by parents to prevent the risks arising from smartphone addiction.

## Figures and Tables

**Table 1 ijerph-19-11600-t001:** General characteristics of the categorized smartphone use group.

Variables		Total		General User Group		Potential-Risk User Group		High-Risk User Group		
		*N*	*%*	*N*	*%*	*N*	*%*	*N*	*%*	*p*-Value
Total		54,948	100.0	41,173	74.9	12,142	22.1	1633	3.0	
Sex										<0.0001
	Male	28,353	51.6	22,521	54.7	5239	43.1	593	36.3	
	Female	26,595	48.4	18,652	45.3	6903	56.9	1040	63.7	
Grade										<0.0001
	High school	28,961	52.7	22,096	53.7	6067	50.0	798	48.9	
	Middle school	25,987	47.3	19,077	46.3	6075	50.0	835	51.1	
Academic achievement										<0.0001
	High	18,217	33.2	12,634	30.7	4778	39.4	805	49.3	
	Moderate	16,585	30.2	12,679	30.8	3531	29.1	375	23.0	
	Low	20,146	36.7	15,860	38.5	3833	31.6	453	27.7	
Household income										<0.0001
	High	7212	13.1	5047	12.3	1859	15.3	306	18.7	
	Average	26,397	48.0	19,650	47.7	5985	49.3	762	46.7	
	Low	21,339	38.8	16,476	40.0	4298	35.4	565	34.6	
Residential type										<0.0001
	Lives with parents	52,332	95.2	39,126	95.0	11,651	96.0	1555	95.2	
	Lives without parents	2616	4.8	2047	5.0	491	4.0	78	4.8	
Sleep satisfaction										<0.0001
	Satisfied	16,824	30.6	13,759	33.4	2750	22.6	315	19.3	
	Moderate	18,656	34.0	14,165	34.4	4081	33.6	410	25.1	
	Unsatisfied	19,468	35.4	13,249	32.2	5311	43.7	908	55.6	
Fast food										<0.0001
	≤2 times/week	41,292	75.1	31,903	77.5	8396	69.1	993	60.8	
	≥3 times/week	13,656	24.9	9270	22.5	3746	30.9	640	39.2	
Physical activity										<0.0001
	≤2 days a week	37,241	67.8	26,858	65.2	9120	75.1	1263	77.3	
	≥3 days a week	17,707	32.2	14,315	34.8	3022	24.9	370	22.7	
Lifetime smoking experience										<0.0001
	Never	48,867	88.9	37,062	90.0	10,527	86.7	1278	78.3	
	Ever	6081	11.1	4111	10.0	1615	13.3	355	21.7	
Alcohol consumption										<0.0001
	Never	49,056	89.3	37,181	90.3	10,558	87.0	1317	80.6	
	Ever	5892	10.7	3992	9.7	1584	13.0	316	19.4	

**Table 2 ijerph-19-11600-t002:** Bivariate association between smartphone addiction and suicide ideation and suicide attempt of study population.

Variables		Total		Suicide Ideation			Suicide Attempt		
				Yes		*p*-Value	Yes		*p*-Value
		*N*	*%*	*N*	*%*		*N*	*%*	
Total		54,948	100.0	5979	10.9		1121	2.0	
Smartphone addiction						<0.0001			<0.0001
	General user group	41,173	74.9	3680	6.7		721	1.3	
	Potential-risk user group	12,142	22.1	1860	3.4		305	0.6	
	High-risk user group	1633	3.0	439	0.8		95	0.2	
Sex						<0.0001			<0.0001
	Male	28,353	51.6	2254	4.1		382	0.7	
	Female	26,595	48.4	3725	6.8		739	1.3	
Grade						<0.0001			<0.9439
	High school	28,961	52.7	3013	5.5		592	1.1	
	Middle school	25,987	47.3	2966	5.4		529	1.0	
Academic achievement						<0.0001			<0.0001
	High	18,217	33.2	2486	4.5		531	1.0	
	Moderate	16,585	30.2	1551	2.8		271	0.5	
	Low	20,146	36.7	1942	3.5		319	0.6	
Household income						<0.0001			<0.0001
	High	7212	13.1	1332	2.4		292	0.5	
	Average	26,397	48.0	2639	4.8		447	0.8	
	Low	21,339	38.8	2008	3.7		382	0.7	
Residential type						<0.0001			<0.0001
	Lives with parents	52,332	95.2	5594	10.2		1011	1.8	
	Lives without parents	2616	4.8	385	0.7		110	0.2	
Sleep satisfaction						<0.0001			<0.0001
	Satisfied	16,824	30.6	974	1.8		188	0.3	
	Moderate	18,656	34.0	1788	3.3		302	0.5	
	Unsatisfied	19,468	35.4	3217	5.9		631	1.1	
Fast food						<0.0001			<0.0024
	≤2 times/week	41,292	75.1	4268	7.8		799	1.5	
	≥3 times/week	13,656	24.9	1711	3.1		322	0.6	
Physical activity						<0.0066			<0.0403
	≤2 days a week	37,241	67.8	4145	7.5		728	1.3	
	≥3 days a week	17,707	32.2	1834	3.3		393	0.7	
Lifetime smoking experience						<0.0001			<0.0001
	Never	48,867	88.9	4773	8.7		769	1.4	
	Ever	6081	11.1	1206	2.2		352	0.6	
Alcohol consumption						<0.001			<0.001
	Never	49,056	89.3	4809	8.8		796	1.4	
	Ever	5892	10.7	1170	2.1		325	0.6	

**Table 3 ijerph-19-11600-t003:** Results of multivariate logistic regression analysis for the factors associated between Smartphone addiction risk and suicidal ideation and suicide attempt.

Variables		Suicide Ideation		Suicide Attempt	
		*OR*	*95% CI*	*OR*	*95% CI*
Smartphone addiction					
	General user group	1.00		1.00	
	Potential-risk user group	1.50	1.41–1.60	1.10	0.96–1.27
	High-risk user group	2.49	2.21–2.81	1.87	1.48–2.37
Sex					
	Male				
	Female	1.88	1.77–1.99	2.42	2.12–2.77
Grade					
	High school	1.00		1.00	
	Middle school	1.21	1.15–1.29	1.60	1.41–1.82
Academic achievement					
	High	1.00		1.00	
	Moderate	0.93	0.87–1.00	1.05	0.89–1.24
	Low	1.16	1.08–1.24	1.42	1.22–1.65
Household income					
	High	1.00		1.00	
	Average	1.00	0.94–1.06	0.88	0.76–1.01
	Low	1.78	1.65–1.93	1.71	1.45–2.02
Residential type					
	Lives with parents	1.00		1.00	
	Lives without parents	1.36	1.21–1.53	2.13	1.72–2.63
Sleep satisfaction					
	Satisfied	1.00		1.00	
	Moderate	1.54	1.42–1.68	1.24	1.03–1.50
	Unsatisfied	2.52	2.34–2.73	2.14	1.81–2.54
Fast food					
	≤2 times/week	1.00		1.00	
	≥3 times/week	1.09	1.02–1.16	1.02	0.89–1.17
Physical activity					
	≤2 days a week	1.00		1.00	
	≥3 days a week	0.91	0.85–0.97	0.77	0.67–0.87
Lifetime smoking experience					
	Never	1.00		1.00	
	Ever	1.83	1.69–1.99	2.86	2.45–3.35
Alcohol consumption					
	Never				
	Ever	1.62	1.49–1.76	2.11	1.80–2.47

**Table 4 ijerph-19-11600-t004:** The results of subgroup analysis stratified by are independent variables.

Variables		Suicide Ideation					Suicide Attempt			
		General User Group	Potential-Risk User Group		High-Risk User Group		General User Group	Potential-Risk User Group		High-Risk User Group
		*OR*	*OR*	*95% CI*	*OR*	*95% CI*	*OR*	*OR*	*95% CI*	*95% CI*
Sex										
	Male	1.00	1.49	1.35–1.65	2.75	2.23–3.39	1.00	0.96	0.74–1.25	1.68–3.82
	Female	1.00	1.49	1.38–1.62	2.35	2.03–2.73	1.00	1.15	0.97–1.36	1.22–2.15
Academic achievement										
	High	1.00	1.43	1.30–1.58	2.26	1.91–2.68	1.00	1.08	0.88–1.32	1.28–2.38
	Moderate	1.00	1.57	1.39–1.77	3.03	2.36–3.90	1.00	1.09	0.82–1.45	1.32–3.70
	Low	1.00	1.51	1.35–1.69	2.48	1.96–3.14	1.00	1.13	0.86–1.48	1.17–3.10
Residential type										
	Lives with parents	1.00	1.51	1.42–1.61	2.48	2.19–2.81	1.00	1.11	0.96–1.28	1.34–2.22
	Lives without parents	1.00	1.31	0.99–1.73	2.62	1.56–4.40	1.00	1.10	0.66–1.84	1.68–6.89

**Table 5 ijerph-19-11600-t005:** Association between smartphone addiction and suicide ideation and attempt according to individual component of independent variable.

Variables	Suicide Ideation			Suicide Attempt		
	No	Yes		No	Yes	
	*OR*	*OR*	*95% CI*	*OR*	*OR*	*95% CI*
**Smartphone over dependence statement**						
It fails every time you reduce the smartphone hours	1.00	1.26	1.19–1.33	1.00	0.95	0.83–1.07
It is difficult to control the usage time of smartphone	1.00	1.25	1.18–1.32	1.00	0.88	0.78–1.00
It is difficult to keep proper smartphone usage time	1.00	1.28	1.21–1.36	1.00	0.94	0.83–1.06
It is hard to focus on other things if you have a smartphone next to you	1.00	1.39	1.32–1.47	1.00	1.01	0.89–1.14
Thinking of smartphone does not leave your head	1.00	1.70	1.58–1.83	1.00	1.40	1.20–1.63
I strongly desire to use smartphone	1.00	1.66	1.56–1.77	1.00	1.31	1.14–1.50
I have problems with my health because of smartphone use	1.00	1.57	1.46–1.69	1.00	1.65	1.42–1.90
I have a hard time with my family because of smartphone use	1.00	2.08	1.96–2.21	1.00	1.87	1.65–2.13
Due to the use of smartphones, I have experienced severe conflicts in friends, colleagues, and social relationships	1.00	2.42	2.18–2.69	1.00	2.45	2.03–2.95
Due to the smartphone, there are difficulties in performing tasks (such as study or work)	1.00	1.62	1.53–1.73	1.00	1.35	1.18–1.55

The results were adjusted for other covariates. odds ratio (OR); 95% confidence intervals (95% CI).

## Data Availability

Publicly available datasets were analyzed in this study. This data can be found here: (https://kdca.go.kr/yhs, accessed on 26 September 2021).

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
