# Peer review of "Association between Smartphone Addiction and Suicide"

_ijerph, 2022, doi:10.3390/ijerph191811600_

Round 1

Reviewer 1 Report

Dear Authors,

Congratulations for your research, it is simple, clear and precise. However, as a reviewer, I recommend you to make some small nuances.

First of all, the introduction is very brief. I understand that research should be synthetic, but you should extend the explanation to smartphones. There are few references about smartphone addiction, so I think they should be based on Internet addiction or problematic Internet use. Explain what it is, and above all what social and personal needs are involved. There is also a strong relationship between addiction or problematic internet use and depression, this variable can be very interesting. There are meta-analyses that address these issues, review them and add the information you find.  In the introduction I miss a review of the role of sex in addiction and suicide, add a paragraph on this aspect. Remember that all these new references and new information should appear in the discussion.

The discussion is interesting but you need to counter the information more with other research. Add more references and add to the discussion.

The methodology and results are correct, very clear and precise. 

Congratulations for your work

T

Author Response

We were pleased to have the opportunity to revise our paper. In revising our paper, we have carefully considered your comments and suggestions. As instructed, we have attempted to explain the changes made in reaction to all of the reviewers’ comments. The reviewers’ comments were very helpful overall, and we appreciate the constructive feedback on our original submission. After addressing the issues raised, we feel the quality of the paper has greatly improved and we hope you agree. Our response to each comment is as follows, and we attach a revision note with the highlighted, revised sections of the manuscript. Again, thank you for the valuable and helpful comments.

Reviewer 2 Report

The statistical analyses reported in this paper appear reasonably well done. Their descriptions in the text of the paper can be improved, however. Here are some things to attend to.

The "Result" section should be labeled "Results".

In the second paragraph of this section that begins with "Table 2 shows the association between...", you need to make clear in this sentence and in the title of the table that these are bivariate associations and that the p-values are for the Chi-squared statistics that measure these associations.

Similarly, in the title of Table 3 and in the text, you need to make clear that these are the estimates from the multivariate logistic regression analyses.

In general, all of the paragraphs describing the results of the statistical analyses need careful elaboration and details of the analyses and findings. 

Regarding the statistical analyses, you also need to address the question of whether your analyses found any statistically significant interactions among the regressors.

Author Response

(The authors gave the same response as above.)

Round 2

Reviewer 2 Report

The revisions to this paper have been responsive to the previous review and the manuscript is improved. I have no further suggestions for revision.

Author Response

Dear  Reviewer,

Thank you for your great efforts and hard work in reviewing our manuscript. We also express our gratitude for your meaningful comments.

Once again, we are very thankful for the time and great effort you put into this revising process.

Sincerely Oyuntuya,